# High-Flow Nasal Cannula Oxygen Therapy versus Non-Invasive Ventilation in AIDS Patients with Acute Respiratory Failure: A Randomized Controlled Trial

**DOI:** 10.3390/jcm12041679

**Published:** 2023-02-20

**Authors:** Jingjing Hao, Jingyuan Liu, Lin Pu, Chuansheng Li, Ming Zhang, Jianbo Tan, Hongyu Wang, Ningning Yin, Yao Sun, Yufeng Liu, Hebing Guo, Ang Li

**Affiliations:** Department of Critical Care Medicine, Beijing Ditan Hospital, Capital Medical University, Beijing 100015, China

**Keywords:** acquired immunodeficiency syndrome, acute respiratory failure, high-flow nasal cannula oxygen therapy, non-invasive ventilation

## Abstract

Background: Acute respiratory failure (ARF) remains the most common diagnosis for intensive care unit (ICU) admission in acquired immunodeficiency syndrome (AIDS) patients. Methods: We conducted a single-center, prospective, open-labeled, randomized controlled trial at the ICU, Beijing Ditan Hospital, China. AIDS patients with ARF were enrolled and randomly assigned in a 1:1 ratio to receive either high-flow nasal cannula (HFNC) oxygen therapy or non-invasive ventilation (NIV) immediately after randomization. The primary outcome was the need for endotracheal intubation on day 28. Results: 120 AIDS patients were enrolled and 56 patients in the HFNC group and 57 patients in the NIV group after secondary exclusion. Pneumocystis pneumonia (PCP) was the main etiology for ARF (94.7%). The intubation rates on day 28 were similar to HFNC and NIV (28.6% vs. 35.1%, *p* = 0.457). Kaplan–Meier curves showed no statistical difference in cumulative intubation rates between the two groups (log-rank test 0.401, *p* = 0.527). The number of airway care interventions in the HFNC group was fewer than in the NIV group (6 (5–7) vs. 8 (6–9), *p* < 0.001). The rate of intolerance in the HFNC group was lower than in the NIV group (1.8% vs. 14.0%, *p* = 0.032). The VAS scores of device discomfort in the HFNC group were lower than that in the NIV group at 2 h (4 (4–5) vs. 5 (4–7), *p* = 0.042) and at 24 h (4 (3–4) vs. 4 (3–6), *p* = 0.036). The respiratory rate in the HFNC group was lower than that in the NIV group at 24 h (25 ± 4/min vs. 27 ± 5/min, *p* = 0.041). Conclusions: Among AIDS patients with ARF, there was no statistical significance of the intubation rate between HFNC and NIV. HFNC had better tolerance and device comfort, fewer airway care interventions, and a lower respiratory rate than NIV. Clinical Trial Number: Chictr.org (ChiCTR1900022241).

## 1. Introduction

Acquired immunodeficiency syndrome (AIDS) is caused by human immunodeficiency virus (HIV) infection and AIDS patients are prone to various complications due to immunodeficiency and needing critical care support. In the highly active antiretroviral therapy (HAART) era, the overall life expectancy of HIV-infected patients has markedly increased [1,2], and survival has markedly improved, with an estimated intensive care unit (ICU) survival rate of 60–75% [3]. The leading causes of HIV-related death have shifted from opportunistic infectious diseases to chronic conditions after the introduction of HAART [4,5,6]. Acute respiratory failure (ARF) remains the most common diagnosis for ICU admission [6,7], especially in low- and middle-income countries. Critical care support, especially respiratory support, is necessary, and should be provided for AIDS patients with ARF. Invasive mechanical ventilation (IMV) is often required for severe cases, and the need for IMV is associated with high mortality and prolonged hospital stay [4,8]. Therefore, it is essential and urgent to explore non-invasive ventilation therapy. 

Non-invasive ventilation (NIV) has been used for decades and was recommended in some guidelines as the first-line strategy in treating ARF in immunocompromised patients [9,10]. For AIDS patients with pneumonia, NIV could improve respiratory function and reduce the mortality rate compared to conventional oxygen via a mask [11,12]. High-flow nasal cannula (HFNC) oxygen therapy is a new type of respiratory support system that can supply heated and humidified oxygen at high flow rates through special nasal prongs and be widely used on patients with ARF caused by various diseases. As we all know, there was no proof of the efficacy and safety of HFNC in AIDS patients with ARF. Some studies have compared HFNC and NIV in patients with acute hypoxemia respiratory failure, after extubation, with COPD exacerbation and with immunocompromise [13,14,15,16,17] and confirmed HFNC was better than NIV in comfort and tolerance [13,16,17]. In an observational cohort study involving immunocompromised patients without AIDS patients admitted to ICU for ARF, intubation and mortality rates were lower in patients treated with HFNC alone than with NIV [18]. However, there was no study comparing HFNC and NIV in AIDS patients with ARF. In the present study, we aimed to evaluate the efficacy and safety of HFNC versus NIV in AIDS patients with ARF.

## 2. Materials and Methods

### 2.1. Study Design and Ethical Approval

This study was a single-center, prospective, open-labeled, randomized, controlled trial registered at chictr.org (ChiCTR1900022241, 31 March 2019). This study was conducted at the ICU, Beijing Ditan Hospital, the largest designated tertiary care hospital for HIV/AIDS patients in North China. The Ethics Committee of Beijing Ditan Hospital approved this study (Jingdilunkezi, 2018-005-01), and the study conformed to the Declaration of Helsinki. Informed consent was obtained from each enrolled patient.

### 2.2. Screening of Patients

Consecutive AIDS patients with ARF admitted to the ICU or the Center for Infectious Diseases, Beijing Ditan Hospital, from April 2019 to March 2022, were screened daily. All enrolled patients in this study were inpatients infected with HIV and were considered to have AIDS as defined by the Centers for Disease Control and Prevention classification system for HIV infection [19]. A newly diagnosed HIV infection was diagnosed within two months before ICU admission and had not received HAART. If patients at the Center for Infectious Diseases met the inclusion criteria, they would transfer to the ICU. ARF was defined as the onset of respiratory symptoms within 72 h before enrollment, a ratio of the partial pressure of arterial oxygen to the fraction of inspired oxygen (PaO_2_/ FiO_2_) ≤ 300 mmHg or PaO_2_ ≤ 60 mmHg with partial pressure on air with arterial carbon dioxide (PaCO_2_) ≤ 50 mmHg and symptoms of respiratory distress (tachypnea > 25/min, labored breathing, and dyspnea at rest). In addition to ARF, the inclusion criteria were being between 18 to 70 and being willing to accept endotracheal intubation if needed.

Exclusion criteria were impending cardiopulmonary arrest, a disorder of consciousness, absence of airway protective gag reflex, upper airway obstruction, pregnancy or breastfeeding, other organ failures apart from ARF, consent withdrawal, used immunosuppressant, and enrolment in other research protocols. ARF caused by pneumothorax or massive pleural effusion, acute exacerbation of chronic lung disease, cardiogenic pulmonary edema, and central nervous system lesions were also excluded.

### 2.3. Study Treatments

In the HFNC group (AIRVO™ 2, Fisher & Paykel Healthcare, Auckland, New Zealand), subjects were given suitable large-bore nasal prongs according to the patient’s nostrils. HFNC was used with a heated humidified circuit, and the initial flow was 40 L/min with a FiO_2_ of 100% and was then adjusted according to patient tolerance and maintained a peripheral oxygen saturation (SpO_2_) ≥ 92%.

In the NIV group (Stellar™ 150, Resman, Germany), subjects were set in S/T mode with a standard full-face mask. The initial inspiratory airway pressure (IPAP) was set to 12–14 cmH_2_O, the expiratory airway pressure (EPAP) was set to 8–10 cmH_2_O, and the pressure level was titrated to achieve a measured expiratory tidal volume equal to 6–8 mL/kg of ideal body weight and trigger the NIV device with each inhalation. FiO_2_ was set to maintain SpO_2_ ≥ 92%. Between NIV interventions, oxygen was provided through an oxygen bag mask. NIV would last for at least 6 h every day and 2 h every time.

The patient’s initial respiratory support was targeted to last at least 2 h and then continue as needed. If dyspnea had disappeared, or they had maintained more than 93% SpO_2_ using nasal cannula oxygen with 5 L/min flow rate and respiratory frequency ≤ 25/min, HFNC or NIV were withdrawn. Oxygen was provided through an oxygen bag mask if HFNC or NIV treatment failed due to adverse events but did not meet the criteria for endotracheal intubation. What is more, we did not perform NIV as a rescue therapy if HFNC treatment failed.

The criteria for endotracheal intubation were worsening respiratory distress, defined as severe dyspnea (respiratory rate > 40/min or lack of improvement in signs of high respiratory muscle workload), significant hypercapnia with pH ≤ 7.30, SpO_2_ less than 90% with FiO_2_ greater than 60%, refractory hypoxemia (PaO_2_ < 50 mmHg with sufficient oxygen therapy) and copious tracheal secretions, intolerance to the interfaces, cardiac arrest or obvious hemodynamic instability, and severe disturbances of consciousness.

Both groups’ therapeutic managements, besides respiratory support, were according to current guidelines. Intensivists determined the use of sedation medicine and prone position.

### 2.4. Data Collection

For all included patients, age, gender, body weight, comorbidity, smoking history, and HIV-related data were collected. We also recorded the clinical and laboratory data just before the enrollment and clinical-related data during the treatment and follow-up. The severity of illness was evaluated by the Acute Physiological and Chronic Health Status Score II (APACHE II) within 24 h of ICU admission and the sequential organ failure assessment (SOFA) at inclusion [20,21]. The vital signs (heart rate, systolic blood pressure (SBP), diastolic blood pressure (DBP), respiratory rate), the arterial blood gases (arterial pH, PaCO2, PaO_2_/FiO_2_), and calculated the HACOR score and ROX index at baseline, 2 and 24 h after randomization, were recorded. The HACOR score is based on clinical and laboratory parameters, including heart rate, acidosis (assessed by pH), consciousness (evaluated by Glasgow Coma Scale), oxygenation (assessed by PaO_2_/FiO_2_ ratio), and respiratory rate. The ROX index was calculated by dividing SpO_2_/FiO_2_ to respiratory rate. A visual analogue scale (VAS, 0–10 scoring), on which 0 indicated absence and 10 indicated the highest possible levels, was used to express feelings of dyspnea and device discomfort and completed at the same three points. The cause of ARF was determined by the consensus of three senior intensivists (J.L., L.P., and C.L.). Initial settings and ventilation characteristics were collected at 2 h after randomization during NIV or HFNC treatment. Airway care interventions were defined as times to correct unplanned device displacement or assist in the removal or fixation of the device. Adverse events were nasofacial skin breakdown, intolerance, nasal prongs or masks broken, and air leaks. Intolerance was defined as patient-reported complaints due to airway dryness, claustrophobia, gastric distension, ocular irritation, headache, breathlessness, or device-related discomforts like airflow, pressure, temperature, and noise. The complaints of intolerance for patients who need immediate intubation were not included in this adverse event. All data were collected on a dedicated case report form.

### 2.5. Outcomes

The primary outcome was the proportion of patients who required endotracheal intubation on day 28. Secondary outcomes were day 28 and day 90 mortality, day 28 and day 90 ventilatory support-free days (i.e., days alive without HFNC, NIV, and IMV), duration of ICU and hospital length of stay, airway care interventions, and adverse events. VAS scores for dyspnea and device discomfort, arterial blood gas analyses, and vital signs were also compared at baseline, 2 h, and 24 h after randomization.

### 2.6. Sample Size, Randomization, and Statistical Analysis

Based on previous studies [18], we estimated an intubation rate of 55% in the NIV group and 35% in the HFNC group. A sample size of 47 in each group would have a power of 80% to detect such a difference (with a two-sided α of 0.05). According to our previous clinical experience, some patients would give up treatment before endotracheal intubation and loss to follow-up, as considering the potential 20% dropouts, the total number of patients included was rounded up to 120.

Randomization was performed using a computer-generated randomization sequence based on permuted blocks of four participants, and allocation was concealed through an opaque envelope. When patients met the criteria, they were randomly assigned in a 1:1 ratio to the HFNC group or the NIV group. The baseline was defined as the time of randomization, and respiratory support of HFNC or NIV was started immediately after randomization.

All analyses regarding the primary outcome were performed based on the intention-to-treat principle [22]. Patients who died during the study were assigned scores of 0 for ventilator-free days [23]. Numerical variables were expressed as mean (±standard deviation, SD) or median (interquartile, IQR) according to the result of the Kolmogorov–Smirnov test and were compared using Student’s *t*-test or Mann–Whitney U test. Categorical variables were expressed as numbers (percentages) and were evaluated using the chi-square test or Fisher’s exact probability tests. The comparison of VAS scores for dyspnea and device discomfort, arterial blood gas analyses, and vital signs at baseline, 2 and 24 h for each group was performed by paired sample *t*-test or Wilcoxon tests. Kaplan–Meier curves were plotted to assess the time from enrollment to endotracheal intubation and were compared by the log-rank test. Missing data were sparse and not replaced. A two-sided *p* < 0.05 was considered significant. Data analyses were conducted using IBM SPSS Statistics version 22 (SPSS Inc., Chicago, IL, USA).

## 3. Results

### 3.1. Patient Characteristics

Among 167 AIDS patients with ARF admitted during the study period, 120 patients were randomized to the NIV or HFNC group after 47 patients were excluded for various reasons. No patient was lost to follow-up. Data from 113 patients (56 patients in the HFNC group and 57 patients in the NIV group) were analyzed as seven patients were excluded for consent withdrawal after randomization (Figure 1). The baseline characteristics of enrolled AIDS patients in both groups were well-matched (Table 1). Before randomization, oxygen was provided through a nasal cannula, oxygen bagless mask, or oxygen bag mask with no more than a 10 L/min flow rate. The patients had a respiratory rate of 31 ± 8/min and SpO_2_ of 88% (84–92%) on room air. Overall, 93.8% of the patients were male and the mean (±SD) age was 42 ± 12 years. There were also no significant differences in vital signs, arterial blood gases, and VAS scores of dyspnea and device discomfort between the two groups at baseline (Table 2). The leading cause of acute respiratory failure was pneumocystis pneumonia (PCP), a diagnosis of 107 patients (94.7%), and all patients received corticosteroids as adjuvant treatment. The interval time before ICU admission was 0 (0–2) days.

### 3.2. Treatments

In the HFNC group, the gas flow rate was 40 (35–45) L/min with a FiO_2_ of 60% (50–80%). In the NIV group, the IPAP level was 12 (10–14) cmH_2_O and the EPAP level was 7.5 (6–8) cmH_2_O with FiO_2_ 50% (40–60%), resulting in a tidal volume of 7.6 ± 2.1 mL/kg. NIV was delivered for 8.5 (6.5–10) hours daily. Compared to the details at 2 h after randomization, the details at 24 h had no significant differences between the two groups. The duration was longer with HFNC than NIV (6 (3–8) days vs. 4 (3–6) days, *p* = 0.020). 

### 3.3. Primary Outcome

A total of 36 patients (31.9%) required intubation and 7 patients received intubation within 24 h after randomization. The intubation rate on day 28 in the HFNC group was similar to that in the NIV group (28.6% vs. 35.1%, *p* = 0.457; Table 3). Additionally, Kaplan–Meier curves showed no statistical difference in cumulative intubation rates between the two groups (log-rank test 0.401, *p* = 0.527, Figure 2). In patients who required intubation, the interval between enrollment and intubation was 3.0 ± 2.5 days, and IMV lasted for 260 ± 196 h. The main reason for intubation was respiratory failure, which happened in 109 patients (96.5%; 54 in the HFNC group and 55 in the NIV group). Two patients in the HFNC group had obvious hemodynamic instability; one patient had a cardiac arrest, and the other patient suffered an epileptic seizure in the NIV group. 

### 3.4. Secondary Outcomes

Ventilator-free days on day 28 and day 90, mortality on day 28 and day 90, air leaks, and hospital length of stay had no significant differences between the two groups (all *p* > 0.05, Table 3). The day 28 mortality was 23.0% and the day 90 mortality was 29.2%, including the hospital mortality. The number of airway care interventions in the HFNC group was fewer than in the NIV group (6 (5–7) vs. 8 (6–9), *p* < 0.001). The rate of all adverse events had no significant differences between the two groups (*p* = 0.186). The rate of intolerance in the HFNC group was lower than in the NIV group (1.8% vs. 14.0%, *p* = 0.032). No patient in the two groups stopped treatment due to intolerance after adjusting the settings. There was one case of nasal prongs broken caused by excessive stretching. Air leaks developed in 9 patients, pneumothorax developed in 8 patients (4 in each group), and pneumomediastinum developed in 1 patient in the HFNC group. The ICU length of stay was 9 (6–14) days, and the ICU length of stay was longer in the HFNC group than in the NIV group (11 (8–15) days vs. 7 (6–13) days, *p* = 0.005) Table 3). 

#### VAS Scores, Vital Signs, and Blood Gas Analyses

The VAS scores of device discomfort in the HFNC group were significantly lower than that in the NIV group at 2 h (4 (4–5) vs. 5 (4–7), *p* = 0.042) and at 24 h (4 (3–4) vs. 4 (3–6), *p* = 0.036) after randomization. What is more, the respiratory rate in the HFNC group was significantly lower than that in the NIV group at 24 h (25 ± 4/min vs. 27 ± 5/min, *p* = 0.041). However, there were no significant differences in VAS score of dyspnea, SBP, DBP, heart rate, blood gas analysis parameters, HACOR score, and ROX index at baseline, 2 h and 24 h after randomization between the HFNC group and the NIV group (Table 2). 

The VAS scores for dyspnea and device discomfort and respiratory rate in both groups at 2 h and 24 h after randomization were lower than the baseline levels. Moreover, the levels at 24 h were lower than at 2 h after randomization. The PaO_2_/FiO_2_ and ROX index in both groups at 2 h and 24 h after randomization has higher than the baseline levels. In the NIV group, the HACOR score at 2 h and 24 h after randomization was lower than the baseline level (Table 2).

## 4. Discussion

To the best of our knowledge, this is the first randomized controlled trial to compare the treatment of HFNC and NIV in AIDS patients with ARF. In our study, we included AIDS patients with ARF for two reasons: first, ARF remains the main cause of ICU admission in AIDS patients [6,7]; second, the efficacy and safety of HFNC in AIDS patients with ARF has not been demonstrated so far. HFNC has shown several beneficial effects over NIV in immunocompromised patients with ARF, such as a decrease in the intubation rate and mortality rate [18,24,25], and there are some drawbacks of using NIV, such as the discomfort of the mask, poor patient–ventilator interaction, and so on [26]. We intended to determine whether HFNC had a good performance on clinical outcomes and explore its safety as a first-line therapy for AIDS patients with ARF. In this randomized controlled trial in AIDS patients with ARF, the primary outcome, intubation rate on day 28, and total adverse events showed no significant difference between HFNC and NIV. However, compared with NIV, HFNC had better tolerance and device comfort and fewer airway care interventions. The respiratory rate was significantly lower in the HFNC group than in the NIV group at 24 h. HFNC appears to be an effective way of respiratory support for AIDS patients with ARF.

In immunocompromised critically ill patients with hypoxemic ARF, the intubation rate was 26.1–46% [24,27,28], and IMV occurred within 1 (0–2) day [29]. In our cohort, the intubation rate was 31.9%, and intubation occurred within 3.0 ± 2.5 days. ARF in immunocompromised patients could worsen faster and intubation could occur earlier than in AIDS patients. Our results showed relatively high performances with HFNC and NIV, which may be explained by the homogeneousness of enrolled patients with a critical illness. A multicenter randomized trial in immunosuppression patients with hypoxemic ARF showed that a 2-h trial with HFNC did not reduce the intubation rate [29]. In the HIGH randomized clinical trial, the intubation rate was not significantly different between HFNC and standard oxygen therapy (38.7% vs. 43.8%). However, HFNC had a higher PaO_2_/FIO_2_ and lower respiratory rate after 6 h [28]. The benefit of NIV for acute hypoxemic respiratory failure in immunocompromised patients remains controversial, as the study design and control cohort were different [18,27,30,31]. In a prospective case-control trial in AIDS patients with PCP-related ARF, compared to IMV, the use of NIV avoided 67% of intubation [30]. In a randomized trial conducted among 374 critically ill immunocompromised patients, compared with oxygen therapy, early NIV did not reduce the intubation rate [31]. In a randomized controlled trial, the mortality rate and intubation rate on day 28 of critically ill immunocompromised patients with acute respiratory failure did not differ between HFNC alone and NIV alternating with HFNC [27]. The intubation rate showed no significant difference between HFNC and NIV in this study, different from the result of Cowdrey’s study [18], which may be related to the relatively early admission and enrollment of critically ill patients without any other organ failure. PCP was the main etiology for ARF in this study. HIV-infected PCP was the primary opportunistic infectious disease and usually develops a sub-acute course of disease progression with a higher risk of respiratory failure and mortality. The disease progress in AIDS patients was different from other immunocompromised patients.

For AIDS patients with hypoxemic respiratory failure, oxygenation improved with a progressive sequential elevation of positive end-expiratory pressure (PEEP), and the recommended level of PEEP was about 10 cmH_2_O considering elevations in PaCO_2_ [11]. However, face mask intolerance and air leaks had a higher incidence at high pressure, and impeded oxygen concentration and oxygenation [32]. Considering the pressure, the initial setting of EPAP in this study was 8–10 cmH_2_O higher than in other studies [14,15,18,33,34]. HFNC can produce a PEEP effect, and pharyngeal pressure would remain no more than 3 cmH_2_O even at 60 L/min flow [35]. As the setting of EPAP in the NIV group was higher than PEEP produced by high flow in HFNC, oxygenation improvement should be better with NIV. However, there was no significant difference in blood gas values between the two groups at 2 h and 24 h after randomization. We could not demonstrate a statistically significant improvement in oxygenation parameters between HFNC and NIV. The discontinuity of NIV and potential ventilator-induced lung injury may influence the oxygenation effect. The excellent tolerance of HFNC and the increase in effective alveolar ventilation caused by the washout effect of dead space in HFNC may gradually make up for the above deficiency. The lower respiratory rate in the HFNC group than the NIV group was also confirmed in a multicenter, randomized controlled trial for chronic obstructive pulmonary disease patients after extubation, which may be related to the relatively higher FiO_2_ [15].

The number of airway care interventions, intolerance, and the VAS score of device discomfort in the HFNC group were also significantly lower than those in the NIV group, which may be related to the HFNC nasal prongs design and better comfort. Like in this study, many studies have found that HFNC is often better tolerated than NIV [14,15,26,27,31,34,36], but data on AIDS patients with ARF has been limited. Airway dryness and discomfort were significantly lower in the HFNC group [37]. Patients with NIV masks usually need to remove or displace their masks due to drinking and eating, communication, sputum clearance, discomfort, or intolerance, which would significantly increase the interventions of airway care [38]. Patients with HFNC were at least not restricted in eating, drinking, and communicating, and the incidence of breakdown and displacement of nasal prongs was extremely low in HFNC, which related to the lower rate of airway care interventions and intolerance and a better feeling of device comfort than NIV.

Our study has some limitations. First, due to the nature of the interventions, attending physicians could not be blinded to the study group since the devices were different. Second, as no selection bias exists in the study, most of our patients were male and PCP was the main etiology for ARF. These factors may limit the external validity of our findings. Third, the statistical power of our study was lower than planned, as the intubation rates in both groups in this study were significantly lower than the intubation rates used for evaluating the sample size.

## 5. Conclusions

In this trial, we could not demonstrate a significant difference in intubation rate among AIDS patients with ARF between HFNC and NIV. HFNC may be a feasible alternative to NIV with better tolerance and device comfort, fewer airway care interventions, and a lower respiratory rate in treating AIDS patients with ARF.

## Figures and Tables

**Figure 1 jcm-12-01679-f001:**
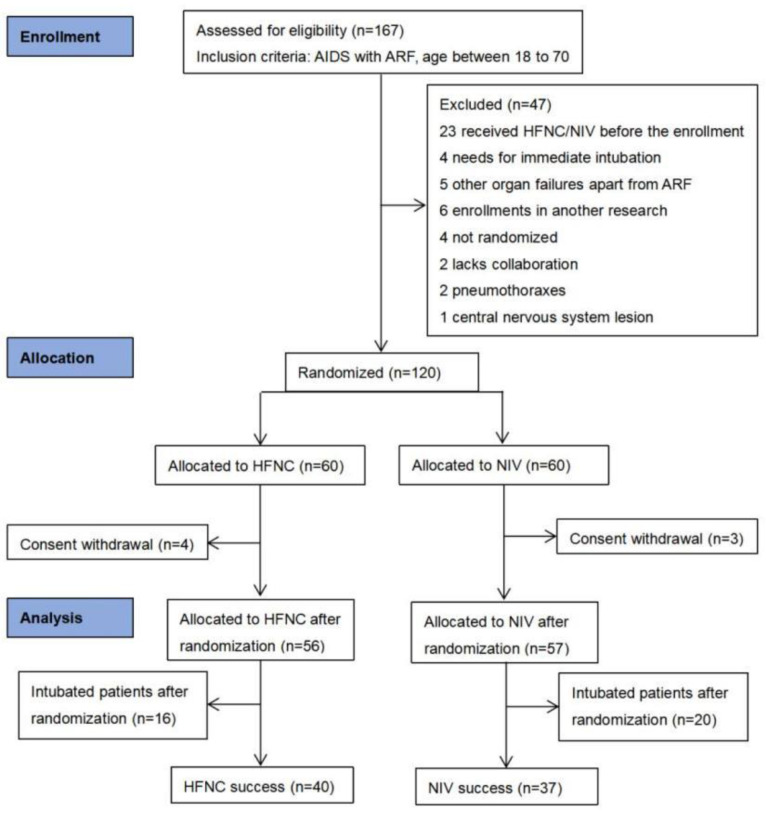
Flow chart of patient enrollment. AIDS, acquired immunodeficiency syndrome; ARF, acute respiratory failure; HFNC, high-flow nasal cannula oxygen therapy; NIV, non-invasive ventilation.

**Figure 2 jcm-12-01679-f002:**
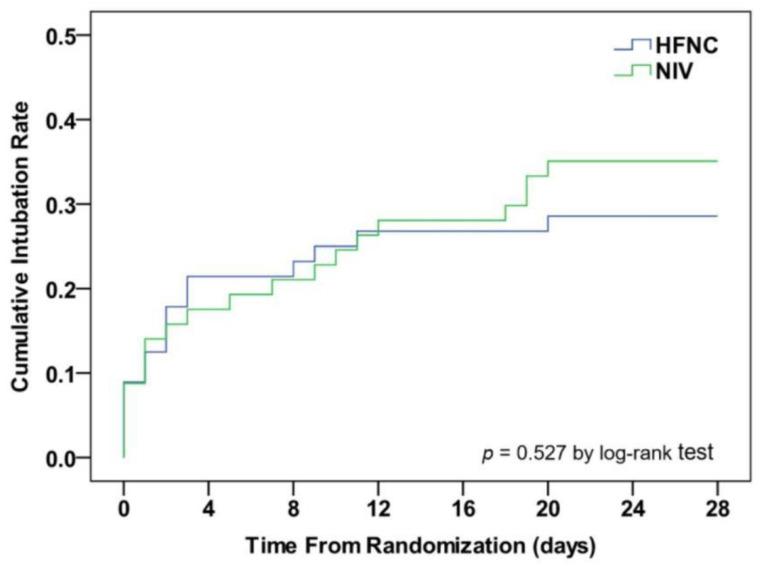
Kaplan–Meier curve analysis of cumulative intubation rate in AIDS patients with ARF. AIDS, acquired immunodeficiency syndrome; ARF, acute respiratory failure; HFNC, high-flow nasal cannula oxygen therapy; NIV, non-invasive ventilation.

**Table 1 jcm-12-01679-t001:** Baseline characteristics of enrolled AIDS patients.

	HFNC Group (*n* = 56)	NIV Group (*n* = 57)	*p* Value
Age (years)	41 ± 12	39 ± 12	0.478
Weight (kg)	62 ± 10	60 ± 12	0.185
Male	54 (96.4%)	52 (91.2%)	0.438
Smoking history	12 (21.4%)	13 (22.8%)	0.860
Comorbidities			0.810
Diabetes mellitus	4 (7.1%)	3 (5.3%)	
Hypertension	3 (5.4%)	3 (5.3%)	
Coronary artery disease	0 (0)	1 (1.8%)	
Chronic kidney disease	0 (0)	1 (1.8%)	
HIV-related data			
Newly diagnosed HIV infection	41 (73.2%)	41 (71.9%)	0.878
On HAART before admission	13 (23.2%)	13 (22.8%)	0.959
CD4 (cells/μL)	16 (6–32)	14 (8–26)	0.947
HIV viral load (copies/mL) ^†^	168,760 (72,472–394,631)	135,992 (70,661–304,846)	0.651
Clinical and laboratory data			
Cough	42 (75.0%)	43 (75.4%)	0.957
Abnormal breath sound	26 (46.4%)	30 (52.6%)	0.510
APACHE II score	14 (11–18)	14 (10–17)	0.429
SOFA score	3 (2–3)	3 (2–4)	0.108
Respiratory rate (per min)	31 ± 7	31 ± 10	0.939
SpO_2_ (%) ^‡^	87 (83–90)	88 (85–90)	0.863
PaO_2_ (mmHg) ^‡^	56 (48–60)	57 (51–62)	0.266
Lactate dehydrogenase (U/L)	453 (354–572)	446 (301–67)	0.503
Albumin (g/dL)	29.5 (26.8–32.5)	30.9 (27.1–34.0)	0.291
Serum 1,3-d-glucan (pg/mL)	63 (21–172)	80 (20–231)	0.641
Cause of ARF			0.513
PCP	53 (94.6%)	54 (94.7%)	
Pulmonary tuberculosis	2 (3.6%)	2 (3.5%)	
Bacterial pneumonia	1 (1.8%)	1 (1.8%)	
Time before ICU (days)	0 (0–2)	0 (0–2)	1.000

Abbreviations: AIDS, acquired immunodeficiency syndrome; HFNC, high-flow nasal cannula oxygen therapy; NIV, non-invasive ventilation; HIV, human immunodeficiency virus; HAART, highly active antiretroviral therapy; APACHE II, Acute Physiology and Chronic Health Evaluation II; SOFA, Sequential Organ Failure Assessment; SpO_2_, peripheral capillary oxygen saturation; PaO_2_, arterial oxygen partial pressure; ARF, acute respiratory failure; PCP, pneumocystis pneumonia. ^†^ 102 patients had HIV viral load, 54 in the HFNC group, and 48 in the NIV group. ^‡^ SpO_2_ and PaO_2_ were obtained on air.

**Table 2 jcm-12-01679-t002:** VAS scores, vital signs, and arterial blood gas analyses of enrolled AIDS patients.

	HFNC Group (*n* = 56)			NIV Group (*n* = 57)		
	Baseline	At 2 h	At 24 h ^†^	Baseline	At 2 h	At 24 h ^†^
VAS score for dyspnea ^‡^	6 (5–7)	6 (5–7) *	5 (2–6) *^,^**	6 (5–7)	5 (4–7) *	5 (3–8) *^,^**
VAS score for device discomfort ^‡^	5 (5–7)	4 (4–5) *^,^***	4 (3–4) *^,^**^,^***	5 (5–7)	5 (4–7) *	4 (3–6) *^,^**
SBP (mmHg)	116 ± 13	114 ± 14	109 ± 9	117 ± 16	113 ± 13	111 ± 13
DBP (mmHg)	71 ± 10	69 ± 9	70 ± 9	73 ± 12	70 ± 9	71 ± 8
Heart rate (per min)	100 ± 16	98 ± 19	85 ± 16 *	106 ± 23	98 ± 19	88 ± 15 *
Respiratory rate (per min)	31 ± 7	28 ± 5 *	25 ± 4 *^,^**^,^***	32 ± 9	30 ± 6 *	27 ± 5 *^,^**
Arterial pH	7.45 (7.42–7.48)	7.45 (7.42–7.48)	7.43 (7.42–7.48)	7.45 (7.42–7.46)	7.44 (7.42–7.46)	7.44 (7.42–7.47)
PaCO_2_ (mmHg)	31.4 ± 5.5	32.5 ± 7.3	31.3 ± 5.0	32.8 ± 6.3	33.6 ± 6.5	31.2 ± 2.7
PaO_2_/FiO_2_	186 ± 56	202 ± 66 *	211 ± 66 *	177 ± 66	198 ± 71 *	203 ± 42 *
HACOR score	2 (0.25–5)	2 (0–4.25)	2 (0–5)	4 (0–6)	2 (0, 5) *	2 (0–5) *
ROX index	5.11 (4.22–6.32)	7.27 (4.55–12.50) *	7.08 (4.53–12.73) *	5.64 (4.13–8.55)	7.58 (5.07–13.42) *	7.58 (5.13–14.63) *

Abbreviations: VAS, visual analogue scale; AIDS, acquired immunodeficiency syndrome; HFNC, high-flow nasal cannula oxygen therapy; NIV, non-invasive ventilation; SBP, Systolic blood pressure; DBP, Diastolic blood pressure; PaCO_2_, partial pressure of arterial carbon dioxide; PaO_2_, partial pressure of arterial oxygen; FiO_2_, fraction of inspiration oxygen. ^†^ Outcome was evaluated on patients still receiving the assigned treatment at 24 h (52 in HFNC and 54 in NIV). ^‡^ VAS score on which 0 indicated absence and 10 indicated the highest possible levels of dyspnea and device discomfort. * Compared with the baseline value in the same group, *p* < 0.05. ** Compared with 2 h value in the same group, *p* < 0.05. *** Compared with NIV at the same time point, *p* < 0.05.

**Table 3 jcm-12-01679-t003:** Outcomes and cause analyses of enrolled AIDS patients.

	HFNC Group (*n* = 56)	NIV Group (*n* = 57)	Odds Ratio (95% CI)	*p* Value
Intubation on day 28	16 (28.6%)	20 (35.1%)	0.740 (0.334 to 1.639)	0.457
Interval between enrollment and intubation (days) ^†^	2.9 ± 2.5	3.1 ± 2.5		0.894
Length of IMV (hours) ^†^	300 ± 199	228 ± 193		0.278
Reason for intubation ^†^			-	0.513
Respiratory failure	54 (96.4%)	55 (96.5%)	0.982 (0.133 to 7.223)	1.000
Circulatory failure	2 (3.6%)	1 (1.8%)	2.074 (0.183 to 23.545)	0.618
Neurologic failure	0 (0)	1 (1.8%)	-	1.000
Ventilator-free days on day 28	15.9 ± 8.7	15.9 ± 10.2		0.999
Ventilator-free days on day 90	58.4 ± 37.4	58.6 ± 38.7		0.969
Mortality on day 28	11 (19.6%)	15 (26.3%)	0.684 (0.283 to 1.657)	0.399
Mortality on day 90	16 (28.6%)	17 (29.8%)	0.941 (0.418 to 2.118)	0.884
Airway care interventions	6 (5–7)	8 (6–9)		<0.001
Adverse events	10 (17.9%)	17 (29.8%)	0.512 (0.210 to 1.244)	0.186
Nasofacial skin breakdown	3 (5.4%)	5 (8.8%)	0.589 (0.134 to 2.590)	0.716
Intolerance ^‡^	1 (1.8%)	8 (14.0%)	0.111 (0.013 to 0.922)	0.032
Nasal prongs broken	1 (1.8%)	0 (0)	-	0.496
Air leaks	5 (8.9%)	4 (7.0%)	1.299 (0.330 to 5.111)	0.742
ICU length of stay (days)	11 (8–15)	7 (6–13)		0.005
Hospital length of stay (days)	21 (16–33)	25 (13–34)		0.669

Abbreviations: AIDS, acquired immunodeficiency syndrome; HFNC, high-flow nasal cannula oxygen therapy; NIV, non-invasive ventilation; IMV, invasive mechanical ventilation; ICU, intensive care unit. ^†^ The values of intubation and IMV include data for the 36 intubated patients in the overall population. ^‡^ Intolerance was defined as a patient-reported complaint that did not cause treatment interruption.

## Data Availability

The dataset consisting of de-identified participants’ data is available from the corresponding author upon reasonable request.

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
