# Peer review of "High-Flow Nasal Cannula Oxygen Therapy versus Non-Invasive Ventilation in AIDS Patients with Acute Respiratory Failure: A Randomized Controlled Trial"

_jcm, 2023, doi:10.3390/jcm12041679_

Round 1

Reviewer 1 Report

Excellent work. I find some things that could be improved:

1) HACOR score and ROX score could be calculated for study patients.

2) NIV was not performed in HFNC group? This must be clearly stated.

3) NIV was performed in a non continuous way, that means that patients were allowed to rest during hours with which kind of modality ¿HFNC?. This must be cautiously explained as well as the fact that removing "pressure" from these patients during the first 24-48h (look at the time patients were intubated from randomization, 3 days!!!!) can modify the benefitial effect.

Reviewer 2 Report

The authors report the results of a RCT comparing  HFNC and NIV in AIDS patients with acute respiratory failure. My comments are as follows :

Page 2 line 83 inclusion criteria : undergoing endotracheal intubation ??

Page 3 line 101 : During NIV interventions ?? rather between NIV interventions ?

In table 3 intolerance was defined as complaint of the patient that did not cause interruption of treatment. Please detail if for some patients in the two groups intolerance led to stop treatment, even to intubation.

Discussion : two important and recent studies in immunocompromised patients need to be discussed and added in the references : the study by Coudroy published in Lancet Respir Med 2022 which compared HFNC alone with HFNC alternating with NIV ; and the HIGH RCT by Azoulay published in JAMA 2018 which compared HFNC with standard oxygen.
